# A Remote Monitoring System for Rodent Infestation Based on LoRaWAN

**DOI:** 10.3390/s23094185

**Published:** 2023-04-22

**Authors:** Shin-Chi Lai, Szu-Ting Wang, Kuan-Lin Liu, Chang-Yu Wu

**Affiliations:** 1Department of Automation Engineering, National Formosa University, Huwei 632301, Taiwan; shivan0111@nfu.edu.tw (S.-C.L.); 11157119@nfu.edu.tw (C.-Y.W.); 2Smart Machinery and Intelligent Manufacturing Research Center, National Formosa University, Yunlin 632301, Taiwan; 3Doctor’s Program of Smart Industry Technology Research and Design, National Formosa University, Huwei 632301, Taiwan; 4Department of Electronic Engineering, National Yunlin University of Science and Technology, Douliu 64002, Taiwan; m11113019@gemail.yuntech.edu.tw

**Keywords:** Internet of Things, LoRa, LoRaWAN, rodent monitor, remote monitor

## Abstract

Rodent infestations are a common problem that can result in several issues, including diseases, damage to property, and crop loss. Conventional methods of controlling rodent infestations often involve using mousetraps and applying rodenticides manually, leading to high manpower expenses and environmental pollution. To address this issue, we introduce a system for remotely monitoring rodent infestations using Internet of Things (IoT) nodes equipped with Long Range (LoRa) modules. The sensing nodes wirelessly transmit data related to rodent activity to a cloud server, enabling the server to provide real-time information. Additionally, this approach involves using images to auxiliary detect rodent activity in various buildings. By capturing images of rodents and analyzing their behavior, we can gain insight into their movement patterns and activity levels. By visualizing the recorded information from multiple nodes, rodent control personnel can analyze and address infestations more efficiently. Through the digital and quantitative sensing technology proposed at this stage, it can serve as a new objective indicator before and after the implementation of medication or other prevention and control methods. The hardware cost for the proposed system is approximately USD 43 for one sensor module and USD 17 for one data collection gateway (DCG). We also evaluated the power consumption of the sensor module and found that the 3.7 V 18,650 Li-ion batteries in series can provide a battery life of two weeks. The proposed system can be combined with rodent control strategies and applied in real-world scenarios such as restaurants and factories to evaluate its performance.

## 1. Introduction

Rodents have a close relationship with human life as they are widespread and exist in large numbers, causing disease, damage, and loss of harvest. Specifically, rodents can transmit diseases, such as the plague, which can be transmitted from person to person when Yersinia pestis causes pneumonia [1]. Throughout history, there have been three major pandemics of plague recorded in the 6th, 14th, and 18th centuries [2], with rats being primarily connected to the Black Death, which killed 40% of the European population in the 14th century [3]. Additionally, rodents can cause significant crop damage during outbreaks, resulting in crop failure or reduced quality. For example, Norway rats (Rattus norvegicus) and black rats (Rattus rattus) have caused persistent losses in annual rice production, which could supply over 1.8 billion people yearly [4].

As rat footprints are widespread in the worldwide, rodent control is becoming modulated more frequently. There are a few methods to kill rodents including trapping, fumigating, and shooting which intentionally cause diseases [5]. These methods can be generally classified into two types: poisonous (bait and fumigant poisons) and non-poisonous measures [6]. Poisonous measures have had a devastating impact on natural ecosystems, disrupting the balance of populations of native animals and plants [7]. Bait poisons, particularly anticoagulants, are known for their quick-acting and high toxicity to all life forms. Anticoagulants inhibit the production of clotting agents in the rodent’s liver [8], resulting in death due to internal bleeding [9]. Fumigant poisons, such as carbon dioxide, cyanide gas, and aluminum phosphide, are gases used in rodent control. Fumigation has two main advantages over bait stations: firstly, secondary poisoning risks are greatly reduced, and secondly, dependent offspring are killed along with their parent, rather than being left in the mouse nest to die [6]. Non-poisonous measures for rodent control include chemosterilants [10] and traps [11,12,13]. Chemosterilants, such as synthetic steroids, mainly affect the fertility of rodents, but these drugs may also have consequences for other species [10]. However, the use of traps is often labor-intensive [11], and it can lead to the transmission of pathogens during the process of handling dead rodents, although traps can easily collect dead bodies [12]. In recent years, several automated traps have been developed. For example, a sensor with two infrared breakbeams will be triggered if a rodent enters a trap. The trap then closes, and CO_2_ is released to make the rodent asphyxiate [13]. The advantage of non-poisonous measures is that they do not cause any toxicity into the food chain. However, the disadvantage of traps is that they are ineffective if rodents do not enter them [14].

Both urban and rural areas are affected by rat infestations [15]. Consequently, there is a need to track and understand these infestations, which requires a Long Range (LoRa) network for monitoring. With the growing popularity of the Internet of Things (IoT), a Low-Power Wide-Area Network (LPWAN) would be a feasible solution, as it can cover a wide area with a low data rate [16]. Several LPWAN technologies are already available in the market, including Narrowband IoT (NB-IoT), Sigfox, and Long-Range Wide-Area Network (LoRaWAN) [17]. One of the most attractive technologies is LoRaWAN due to its low-power, long-range communication, scalable bandwidth, and reliable robustness [18].

Referring to IoT, wireless sensor networks (WSNs) are widely used for remote sensing and data collection. Table 1 reviews a comparison of different network technologies for IoT in [19,20]. Traditional short-range communication technologies, such as Bluetooth and WiFi, as well as long-range ones, such as cellular, are not suitable for IoT deployments in monitoring applications [21]. Cellular technologies can handle high data rates for transmission traffic and allow for user equipment to connect through cells, each of which includes a base station with a wide bandwidth. However, scaling up operations increases maintenance costs and energy expenditure [22]. WiFi and Bluetooth were designed to handle a higher amount of data, resulting in a large power consumption. As a result, both WiFi and Bluetooth are impractical for battery-operated devices in rural areas. To mitigate energy consumption, lower-energy communication technologies based on IEEE 802.15.4, such as ZigBee [23], have been introduced. However, the mesh network will only be operational within a distance of 80 m between each end-device and router, which poses a challenge if the end-device is dynamically relocated outside the acceptable transmission distance. In summary, short-range wireless communication technologies are not suitable for long-range transmission scenarios. Although cellular communication can provide broad coverage, it increases operating costs and energy expenditure. To address this issue, LPWANs have been developed as wireless communication networks specifically designed for the transmission of small packets through gateways, while minimizing energy consumption [20]. Among the various LPWAN technologies, LoRa stands out for its ability to significantly reduce both energy and maintenance costs. Figure 1 provides a visual comparison of various wireless communication technologies based on several metrics. The choice of communication technology depends on various factors related to the application. It is evident that LoRa has a longer communication range with a lower data rate, making it ideal for monitoring systems. In addition, it is worth noting that the WSN techniques can not only function independently but can also be combined with other techniques. For example, a comparative study between LoRaWAN and Cellular NB-IoT networks for IoT deployment in developing nations was conducted in article [24]. The study demonstrated that while LoRaWAN had superior coverage and battery life, it had lower data rates, whereas Cellular NB-IoT had higher data rates but limited battery life and coverage.

Traditional methods of pest control, such as the use of pesticides, rely solely on the experience of pest control personnel for their effectiveness. To achieve digital and quantitative pest control, a remote monitoring system with LoRa technology must be developed to measure rodent activity and distribution before and after the implementation of pest control strategies. By deploying sensor nodes to collect objective analysis data within a specific area, the monitoring system can serve as evidence for the effectiveness of pest control strategies. This paper introduces a system for remotely monitoring rodent infestations, which reduces the demand for manual labor intensity. Each sensor consists of a small, low-power electronic infrared obstacle detection module, referred to as an IoT node, equipped with a LoRa module to transmit rodent infestation data via LoRa gateways and WiFi to a cloud server. By visualizing the analyzed information across multiple sensing nodes, users can receive suggestions from rodent control experts, who can provide concise insights for responding to rodent infestations. The proposed system not only reduces the need for labor intensity to observe if an area suffers from rodent infestations but also provides real-time information from each sensor.

This paper is organized as follows: Section 2 presents the background of LoRa and LoRaWAN technology and reviews some studies based on LoRaWAN. Subsequently, the system design and results are explained in Section 3 and Section 4, respectively. Future work and conclusions are then addressed in Section 5.

## 2. Background

### 2.1. LoRa and LoRaWAN

LoRa is a wide-area radio communication technique that was developed by Semtech in 2014 [25]. Semtech patented the LoRa modulation and developed various products, such as the LoRa module SX1278. In this approach, we adopted this LoRa module to be the kernel component, and Table 2 provides more detailed specifications for the LoRa module SX1278. Here, are some of the benefits of utilizing a LoRaWAN network for IoT applications:Long-range communication and low power consumption: LoRaWAN technology can provide coverage of up to 15 km in rural areas and up to 5 km in urban areas, while still maintaining low power consumption, allowing for battery-powered devices to operate for several years [26].Secure communication: LoRaWAN uses AES-128 encryption to secure communication between end-devices and gateways, ensuring data privacy and security [27].Scalability and cost-effectiveness: A LoRaWAN network can support thousands of end-devices, making it highly scalable for large-scale IoT deployments. Additionally, the cost of deploying a LoRaWAN network is relatively low due to its low power consumption and the availability of low-cost devices [28].
sensors-23-04185-t002_Table 2Table 2Specification of LoRa module SX1278.ParameterSpecificationOperating VoltageDC 1.8 V–3.7 VFrequency Range137–525 MHzRF Input Level+10 dBmModulationFSK/OOK/LoRaTM/GMSK/MSKBandwidth7.8–500 kHzMaximum Bit RateUp to 300 kbpsReceiver SensitivityDown to −148 dBmOperating Temperature−40 °C to +85 °CRF Output Power+20 dBmRange3–5 KmDimension20.5 × 15.5 × 2.0 mm


The physical layer modulation of LoRa is derived from Chirp Spread Spectrum (CSS) technology [29]. It operates in 433 MHz, 868 MHz, or 915 MHz Industrial, Scientific, and Medical (ISM) frequency bands, which are region-dependent and license-free. LoRa has three configurable parameters: Spreading Factor (SF), Bandwidth (BW), and Coding Rate (CR), which determine the maximum reachable communication coverage, energy expenditure, and data rate.

SF is related to the duration of the chirp, and it can be set from SF6 to SF12. Larger SF represents longer airtime and slower transmission, while the Signal-to-Noise Ratio (SNR) is gained, resulting in a better range [30]. The data rate varies from 0.3 kb/s to 27 kb/s depending on the SF used [31]. The value of BW represents the range of frequencies for signal transmission. There is a trade-off between data rate and communication coverage, with the most common BW being 125 kHz, 250 kHz, and 500 kHz. If a fast transmission is needed, a higher value of BW is better. The smaller value of BW provides better coverage while taking more time to transmit. CR refers to the bits that are encoded to the packet header, which is protected by forward error correction codes [32]. A smaller value of CR causes more time needed to transmit a packet, and users can have better communication coverage. In addition, it also involves more battery consumption.

LoRaWAN is a communication protocol based on LoRa, which is known as a medium access control (MAC) layer protocol standardized by the LoRa Alliance [33]. A LoRaWAN network consists of four basic elements: end-devices, gateways, a network server, and applications [34], as illustrated in Figure 2. The end-devices, which can be sensors, monitors, controllers, and so on, connect to gateways via LoRaWAN radio. Gateways, acting as intermediate devices, forward messages delivered by end-devices to a network server through an Internet Protocol (IP) data channel. The network server can then inspect the integrity of packets and send them to one of the applications. Once the applications receive the packets, they decrypt them and execute the corresponding process based on the assigned command.

### 2.2. Related Applications with LoRa Technology

LoRaWAN has been widely deployed in wireless sensor networks [35] for various applications such as agriculture [36,37], smart cities [38,39], industrial systems [40], traffic [41], and health-care monitoring [42]. These applications have common characteristics, where only small amounts of data need to be transmitted and monitored. LoRaWAN is more practical for non-real-time monitoring and less feasible for real-time monitoring such as multimedia streaming [31]. Therefore, it is suitable for application cases with unbalanced communication, where the uplink data volume is much greater than the downlink data volume. A performance evaluation of LoRa wireless communication in building environments was also presented in [35], where the communication performance was tested by varying payload size, transmission power, and location of wireless modules. The results indicated that LoRa is appropriate for practical application in smart buildings and extensive monitoring.

In agriculture, sensor nodes are distributed in a wide range of fields for monitoring environmental factors, such as temperature, humidity [43], soil moisture, and leaf wetness [44], to maximize yield and minimize operation expenses. In [43], a LoRa-based tree farm monitoring system was presented, which discussed the impact of main physical layer parameters. In [44], a grape farm monitoring system based on LoRaWAN is introduced, which can transmit data from the sensor nodes to cloud servers. In [45], a water leakage detecting system based on LoRaWAN was proposed, which comprises a pressure sensor and a smart valve in a LoRa node for measuring pressure. Additionally, LoRa and LoRaWAN have been utilized in some health-care solutions. For instance, in [42], a remote point-of-care screening system for urinary tract infection was developed using LoRa/Bluetooth technology. The system’s test results are inspected for correct diagnosis based on color identification with high accuracy. The server collects the measurement sample, which can be viewed by users via an Android mobile app. Furthermore, a prototype bed rail was demonstrated in [46] that could send collected data to a nurse station. If LoRa gateways can cover bed rails mounted with sensors in a hospital, the infrastructure maintenance cost and labor intensity can be minimized. Considering the rodent monitoring application, there is limited rodent control research on using IoT technology in recent years. One of the most common traditional approaches is placing bait stations [47,48], which may contain poisons that could harm pets and even people. In 2019, a low-power remote monitoring system was proposed to transmit data on bait and battery [47]. In 2020, [48] presented a low-power bait station monitoring system Bait combined with computer vision, which can wirelessly report data from bait stations. To ensure the sustainability of the Earth, rodent infestation prevention should be implemented using non-toxic approaches that allow for no delay.

## 3. Proposed System Architecture Design

Figure 3 illustrates the overall block diagram of the proposed system, which includes the sensor node, data collection center, and server. The sensor node detects the presence of rodents using infrared (IR) receivers. An ESP32 camera can be added to capture images of intruders, which can be transmitted to the database via WiFi. The proposed system can use two types of sensors: mask-type sensors and reflective-type sensors, as shown in Figure 4a,b. The mask-type sensor consists of an emitter and a receiver arranged on opposite sides. In the normal state, the emitter transmits an IR signal to the receiver. When an object passes by the emitter, the IR signal is blocked, and the receiver cannot receive the signal, which can be used to detect the entry or exit of rodents. The reflective-type sensor consists of an emitter and a receiver placed in the same direction. When the IR emitter reflects on an object, the adjacent receiver receives the reflected IR and distinguishes whether the object is entering or leaving. The main function of the server is to receive, process, and store data in a database. Finally, users can access and review the collected data through websites or applications. Figure 5 shows how the sensor module is installed in a small, flat box to avoid arousing suspicion in rodents. The sensor module’s overall size is 6.4 × 3.4 × 5.5 cm^3^, and the individual components of the sensor module are introduced in Section 3.1. Figure 5b shows the internal design of the sensor module.

### 3.1. Proposed Hardware Design

The proposed system consists of two main components: the end-device (which includes both the mask- and reflective-type sensors) and the data collection center, which are introduced in the following sections.

The hardware of the sensor node primarily handles data collection, LoRa communication, and power supply functions. The circuit design for the sensor node is depicted in Figure 6, and it comprises four main parts: (1) microcontroller (MCU), (2) communication unit, (3) data collection unit, and (4) power supply unit.

#### 3.1.1. The Circuit Design of Sensor Module

The MCU used in the sensor node is the ATMega328P-PU, which is part of the ATMega328 series produced by Microchip Technology Inc. It features 8-channel 10-bit analog-to-digital (A/D) converters with a conversion time ranging from 13 to 260 μs and a maximum frequency of 20 MHz. The power supply is powered by two 3.7 V 18,650 batteries in series, which are converted by IC-AMS1117-5V. The peripheral circuits use an external 16 MHz quartz oscillator as the clock of the MCU. The Real-Time Clock (RTC) module uses I2C interface for communication, while the LoRa module uses serial peripheral interface (SPI). The IR LED light, relay, IR emitter and receiver are controlled by connecting to digital pins. Moreover, the LoRa module operates at 3.3 V, so another IC, AMS1117-3.3V, is used to regulate the voltage from 5 V.

The LoRa module (SX1278) serves as the communication unit in the proposed hardware design, as shown in Figure 6. The LoRa module is powered by a 3.3 V direct current (DC) supply and operates using transistor–transistor logic (TTL) serial communication. It features four working modes: (1) general mode, (2) communicating mode, (3) sensing mode, and (4) sleeping mode. To conserve battery life, the sensor is set to sleeping mode instead of general mode. The working mode can be adjusted mechanically by manipulating the pins. The power supply unit includes a DC-DC converter and two 3.7 V 18,650 Li-ion batteries connected in series. Two DC-DC converters are used: one for DC 7.4 V to DC 5 V, which is powered by an AMS1117-5V chip, and another for DC 5 V to DC 3.3 V, which is powered by an AMS1117-3.3V chip. As mentioned earlier, there are two types of sensors that can be used in the proposed system. The mask-type sensor features an IR emitter and receiver on opposite sides, as shown in Figure 7. When an object passes by, the IR emitter is blocked, and the IR receiver cannot receive the IR signal. This sensor is suitable for use in wider holes where rodents may pass in and out. The reflective-type sensor, on the other hand, features an IR emitter and receiver on the same side, as shown in Figure 8. The IR receiver is triggered when the IR emitter transmits an IR signal. This type of sensor is suitable for use in more limited spaces.

#### 3.1.2. Proposed Circuit Design for Data Collection Gateways

The ESP32 is used as the main controller for the data collection gateways (DCG) shown in Figure 9, and it is connected to the LoRa module through a serial interface. The LoRa module is responsible for receiving data from the sensor nodes and sending them to the cloud server through a LoRa gateway. The antenna is also used to improve the communication range of the DCG. The power supply for the DCG can be provided either through a micro universal serial bus (USB) cable or an external power supply, as shown in Figure 10. When the USB cable is connected to a computer, the transmitted data can be monitored through the serial port. In suitable areas with good signals, the DCG can be directly powered by an external power supply.

### 3.2. Firmware Design

The flowcharts for the sensor module and DCG module are depicted in Figure 11 and Figure 12, respectively. The MCU remains in deep sleep mode 16 h per day (from 4 a.m. to 8 p.m.) and then turns into working mode 8 h per day (from 8 p.m. to 4 a.m.). The proposed sensor module is capable of retrieving time information from the RTC, and if the current time falls within the working period, the MCU will set the module to operate in working mode; otherwise, it will switch back to sleeping mode.

As the MCU is in working mode, the IR sensors are triggered so that the passing direction could be distinguished by the sequence of the triggered order. As mentioned in Figure 4, if IR sensor A is triggered first, then IR sensor B is triggered, with the passing direction is going out. On the contrary, if IR sensor B is triggered first, followed by the triggering of IR sensor A, it indicates that the passing direction is going in. In this circumstance, where both IR sensors are triggered, the obstacle LED will be lit up. Afterwards, the LoRa module is awakened, and the time is obtained from RTC. The packet is assembled before being sent to the DCG. If the MCU receives an acknowledgment (ACK) from the DCG, the LoRa module will be set to sleeping mode. If the ACK is not received, the MCU will wait for 3 s. While the data are still not transmitted, they will be stored in the Electrically Erasable Programmable Read-Only Memory (EEPROM) until a new packet is to be transmitted.

When the MCU is in sleeping mode, it sets a wakeup time to minimize power consumption. Once the wakeup time is reached, the MCU returns to working mode and remains in that state until it needs to go back to sleeping mode again. Here, DCG is charged by an external power supply to stay ready to receive the packet from sensor nodes at any time. After the packet is received, the packet is disassembled and the data are posted to server via ESP32. To confirm that the data are sent successfully to the server, DCG waits for the requests from the database. The retry time is counted by resending the data again, while the retry time reaches 2 times.

#### 3.2.1. Communication Interface

The MCU (ATmega328P-PU) and LoRa module (SX1278) are deeply sleeping in the daytime to save the power consumption. Additionally, during the nighttime, the MCU will turn on the LoRa module and RTC. The communication interface between MCU and LoRa is deployed by self-defined libraries, where MCU is the master and LoRa is the slave. The format of the self-defined packet includes the location of the sensor, time stamp, and passing status.

#### 3.2.2. LoRa Transmission

The proposed system is deployed by LoRa which is the bridge of transmission between the sensor and DCG module due to its low-power, large-transmission distance. There are three parameters of LoRa transmission which are adjusted, as listed below:SF: The value is set as 12 because the transmission distance between the modules is far and the low data rate is acceptable.Transmit (Tx) Power [49]: This relates to the transmit power of the access point (AP) radio. This is a very important parameter. According to their radio regulations, the maximum value is set to 14 dBm. The value is set as a larger number in an obstructed environment, and vice versa. The allowed maximum output power is 2 Watts (W), which is equivalent to isotropically radiated power (EIRP) (outdoor) and 4W EIRP (indoor).Sleeping mode of LoRa: To save the power consumption, the MCU and LoRa are turned to deep sleeping mode in the daytime, and turned on at nighttime according to rodents’ habit.

### 3.3. Software Design

The overview of software architecture is shown in Figure 13. A screenshot of the monitoring webpage regarding the amount of infestation sensed by different sensor nodes is reported in Figure 14. By conducting the mouse image capture experiment, we tallied the data on the rodent forward (entry action) and backward (exit action). The proposed system deploys Node-RED to be the front-end of the website. The database is built up by XAMPP, which is an open-source server providing a number of functionalities through the package so that the users can view the statistics of the visualized data. Node-RED provides the module to represent live data in a dashboard, which is a useful tool to develop the website. JavaScript is used to perform data validation to ensure that format of the user input is correct. Afterwards, ESP32 sends a power-on self-test (POST) through a hypertext transfer protocol (HTTP) request to store data in the database.

XAMPP is a control panel to activate or deactivate the database and website. MYSQL is utilized to manage the database. The database is manipulated via phpMyAdmin including data insertion and query.

Data insertion: The packet received from DCG is stored in a JavaScript object notation (JSON) file, which contains the location, time stamp, and the passing direction. These data are segmented and stored in the corresponding fields of the database.Data query: Users can view the data via the website and inquire about the data. The function of inquiry consists of total rodent infestation and a comparison of each sensor node so that the users can figure out the infested area.

### 3.4. Auxiliary Design of Rodent Activity Image Acquisition

The proposed system design, as shown in Figure 3, incorporates an ESP32-CAM to assist the sensing nodes in making objective judgments and confirming rodent activity. By recording real-time images, we can perform cross-comparison with the action time of the sensing node to verify the accuracy of the sensing data. The ESP32-CAM has the capability to transmit 2 Mega Pixels streaming information with a resolution of 1920 × 1080 to a camera webserver using WiFi technology. Additionally, the real-time data can be recorded and stored for future use. This image streaming mode can support by providing detailed information such as rodent image, rodent activity, occurring time, and environment status. This implies that if no additional data compression algorithms are used, the data rate per second would be achieved at 187 MBps, i.e., 6.22 MB/frame, with a total of 30 frames per second. If a compatible wireless transmission with Lora is taken into consideration, a lower resolution of 640 × 480 and gray image, resulting in a data rate of 307.2 kBps, would be a suitable choice for practical use. This single image captured mode indicates that a single gray image can be captured every 11 s based on the given data size of 307.2 kB, and then the resulting image file can be further divided into ten partitions for data transmission using Lora technology with a maximum data rate of 31.25 kBps per partition.

In this work, the ESP32-CAM captures the intruders in front of each sensor module. Since the distance between the intruder and the camera is short, a wide-angle lens is applied to capture the full field of view for recording the photos of intruders. To evaluate the real situation of the proposed system, we placed another individual ESP32 camera near the sensor module. When the rat goes through the sensor, the image is captured, as shown in Figure 15. Since the rodents usually appear at nighttime, there is no light at the space of the sensor module. A night vision IR light module, which was needed, is utilized to adjust the ambient light through the lens.

## 4. System Test, Analysis, and Discussion

The sensor module is evaluated in two areas: (1) power efficiency; (2) hardware cost. Additionally, we also discuss the detecting results of rodent activity, and some related issues are briefly described.

### 4.1. Power Efficiency

The sensor module can operate in three distinct modes: sensing mode, communicating mode, and sleeping mode. When operating in sensing mode, the MCU keeps detecting rodent activity through IR technology. When the MCU successfully detects any rodent activity, it will switch to the communicating mode and send relevant information, including the current time stamp, sensor number, field number, and other data to the Lora module for transmission. If the current time falls within the non-working period, the MCU will enter sleeping mode, causing the entire module to go into a low-power state until it is time to wake up again. Considering that rodents are more active at night, we assume that the sensor module will work for 8 h per day, measuring from 8 p.m. to 4 a.m. the following day. The remaining time is considered non-working hours, which are from 4 a.m. to 8 p.m., totaling 16 h.

Table 3 shows the measurement results of the power consumption (PC) in a different mode within a sensor module, and the condition is given that rodents assumingly go through the sensing area 10 times per day. The 3.7 V 18,650 Li-ion batteries in series have a capacity of about 5200 mAh, which can provide approximately 2 weeks of battery life. Alternatively, 18,650 Li-ion batteries are rechargeable, which can be swapped at a 2-week interval. The LoRa configuration parameters are summarized in Table 4. It is worth noting that the proposed sensor module is classified as an end-device belonging to device class A. As communication is always initiated by the end-device, the module is capable of sending an uplink message at any time.

### 4.2. Cost Analysis

The cost of each sensor module is listed in Table 5, and the cost of each DCG module is listed in Table 6. The outer mold is made by Poly Lactic Acid (PLA) materials of 3-dimension printing for the sensor module. The total cost of each sensor module is around USD 43. We assume that a total of three sensor nodes and one DCG are employed in one area to monitor the rodent infestation. Therefore, the total cost for each building in the experiment is USD 146. The time to check the sensor module is 2 weeks, which includes battery changes.

### 4.3. Detecting Results of Rodent Activity

This approach involves using images to auxiliary detect rodent activity in various buildings. By capturing images of rodents and analyzing their behavior, we can gain insight into their movement patterns and activity levels. This allows for us to identify areas that may be particularly prone to infestations, and take steps to mitigate the risk. Regarding the experimental design, we chose to test the proposed monitoring system in two buildings. In each building, we installed three sensors and one DCG module in the possible activity area for rodent monitoring, as illustrated in Figure 16a,b. Figure 16c,e depict images of a rodent preparing to enter one sensing area (forward action) in different buildings, respectively. On the other hand, Figure 16d,f show images of a rodent preparing to exit the sensing area (backward action) in different buildings, respectively.

It is worth noting that Figure 14 and Figure 16a,b display the configuration of three sensing modules in each of buildings 1 and 2. In building 1, sensor 1 detected 16 forward actions and 14 backward actions, indicating that 2 mice passed through the sensor during the month without retracing their steps. Sensor 2 recorded the same number of forward and backward actions, suggesting that the detected mouse activity followed a single path. Sensor 3 showed fewer forward actions but higher backward actions, suggesting that mice may be entering the sensing area from other directions. In building 2, all three sensors reported fewer backward actions, indicating that the rats in the environment may be using multiple activity paths.

### 4.4. Other Related Issues

For computational complexity, the proposed kernel detection method can be viewed as a finite-state machine (FSM). Specifically, it only consists of three states: “only sensing node A detected”, “sensing nodes A&B detected”, and “only sensing node B detected”, for detecting rodent forward or backward action, which change based on the IR results. As a result, there are no additional calculation formulas or mathematical operations involved in the process, making the proposed method relatively low in computational complexity. For security verification, we only add a specific string to the packet transmitted by the front-end sensor to the database, and there is a judgment and comparison when received by the back-end sensor. Currently, the proposed design is intended only to prevent packets that are not generated by our device. In terms of data validation, we have conducted tests only on the actions detected by the sensor modules, and the accuracy for both forward and backward actions is 100%. We have not observed any data errors after successfully transmitting packets through both LoRa and WiFi, not only for the sensor modules but also for DCG. However, we have not specifically tested for packet loss during LoRa transmission.

## 5. Conclusions

This study presents a remotely monitored rodent infestation system based on LoRaWAN technology. The proposed system utilizes sensor and DCG modules to autonomously collect data on rodent activities. It not only provides real-time statistics on an application but also enables rodent controllers to identify the appropriate methods for prevention. Although the proposed system has been able to conduct experiments on the detection of rodent activity, the battery power cannot support the system to remain in sensing mode continuously for 2 weeks, 24 h a day. This means that we can only estimate the rodent activity based on partial activity measurement results. The data obtained from the infrared sensors provide only rough references, and to achieve accurate judgments, it is necessary to rely on ESP32-CAM image captures and the additional development of artificial intelligent (AI) recognition algorithms. Furthermore, the current proposed auxiliary ESP32-CAM image capture relies on a power outlet and cannot be used with battery power. However, we still believe that the developed system can enhance the overall technology for rodent control in the future. Through the digital and quantitative sensing technology proposed at this stage, it can serve as a new objective indicator before and after the implementation of medication or other prevention and control methods. Future research can be conducted by incorporating image recognition technology into the system to identify the types of intruders that the current system does not classify. Additionally, more field testing in both high- and low-density areas in collaboration with pest control enterprises is necessary to verify the system’s effectiveness.

## Figures and Tables

**Figure 1 sensors-23-04185-f001:**
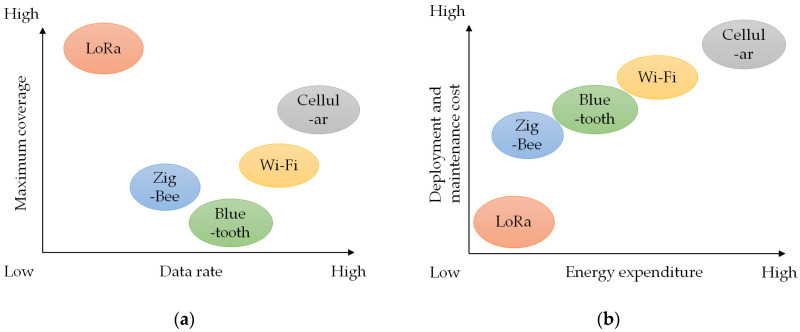
Comparison of various communication technologies in terms of: (**a**) maximum coverage, data rate; (**b**) energy expenditure and operational cost.

**Figure 2 sensors-23-04185-f002:**
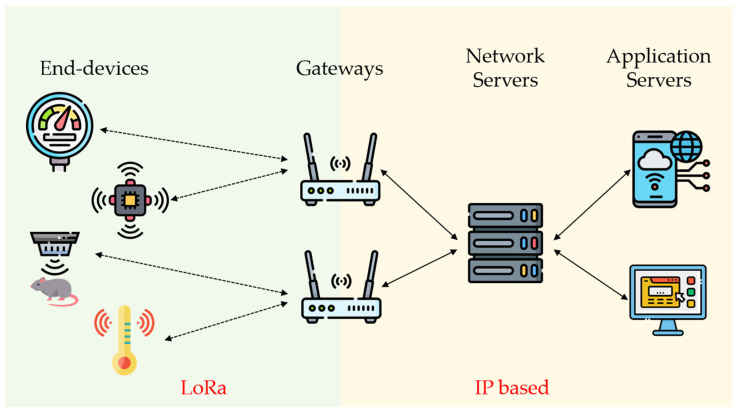
General LoRaWAN network topology.

**Figure 3 sensors-23-04185-f003:**
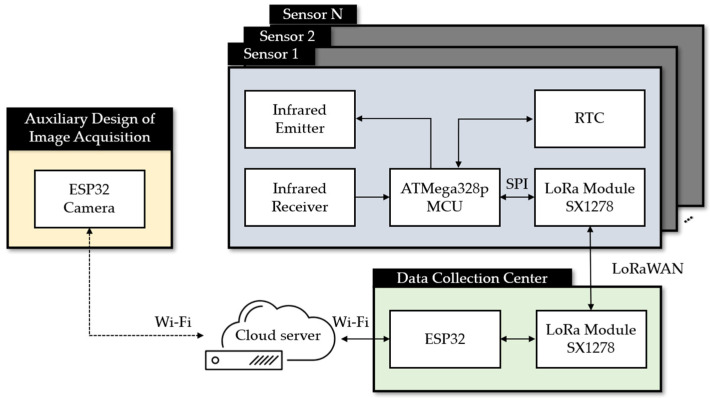
Overall block diagram of the proposed system.

**Figure 4 sensors-23-04185-f004:**
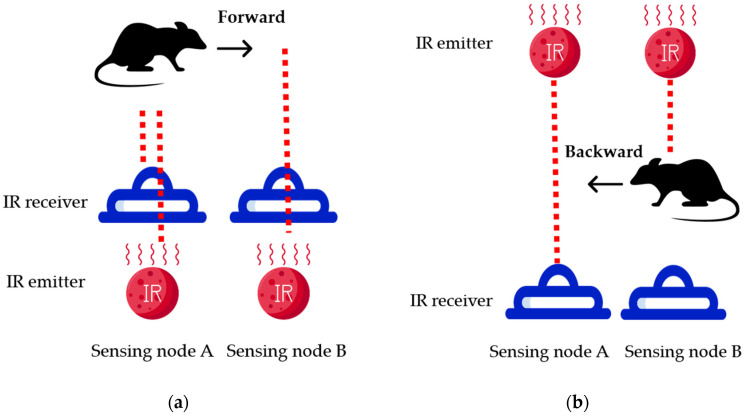
Two types of sensor module: (**a**) mask-type sensor; (**b**) reflective-type sensor.

**Figure 5 sensors-23-04185-f005:**
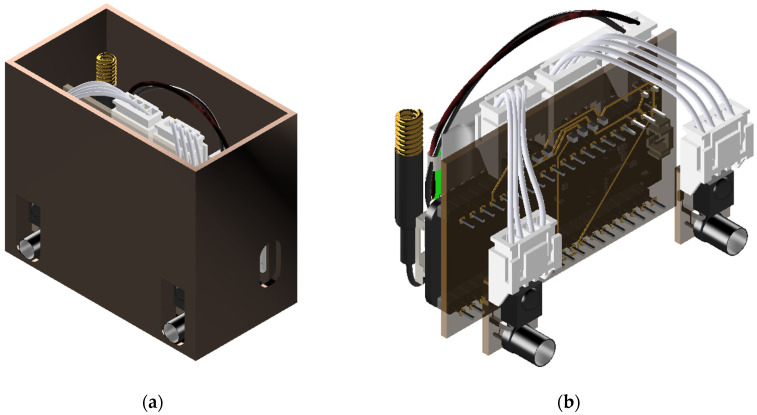
Overall view of the sensor module in (**a**) and view of an internal design in (**b**).

**Figure 6 sensors-23-04185-f006:**
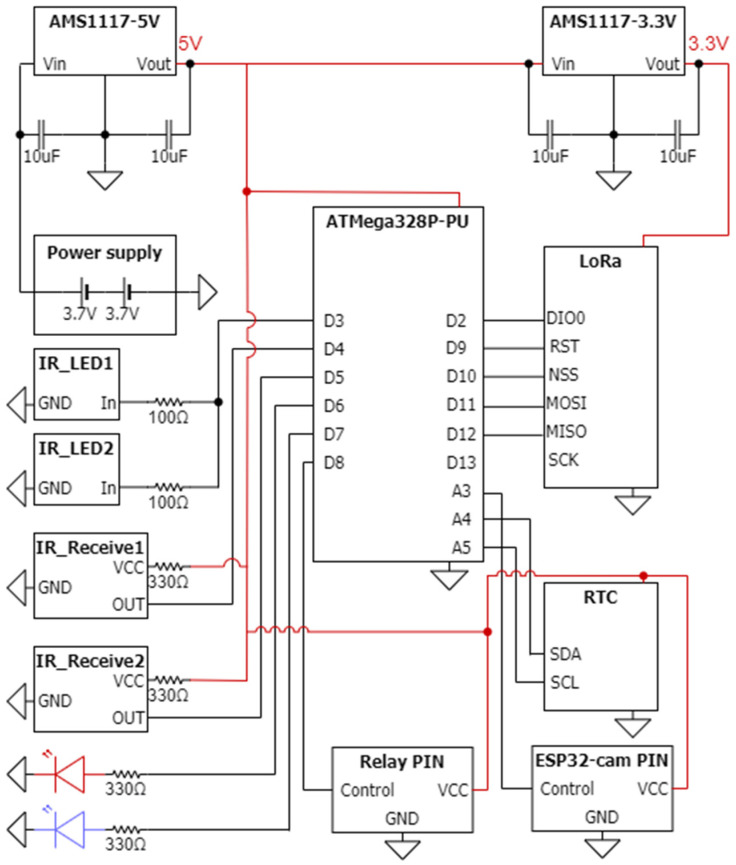
Block diagram of the proposed sensor node with LoRa communication.

**Figure 7 sensors-23-04185-f007:**
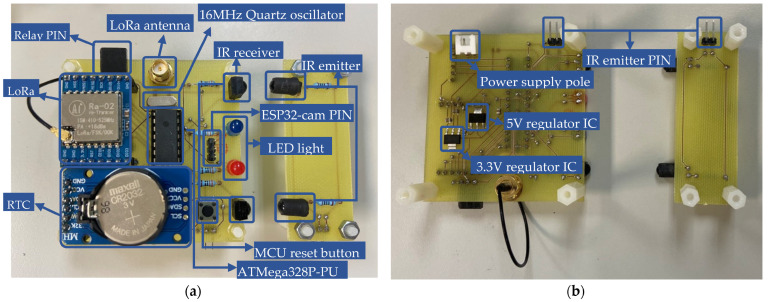
Physical diagram of the mask-type sensor node: (**a**) **top** view; (**b**) **bottom** view.

**Figure 8 sensors-23-04185-f008:**
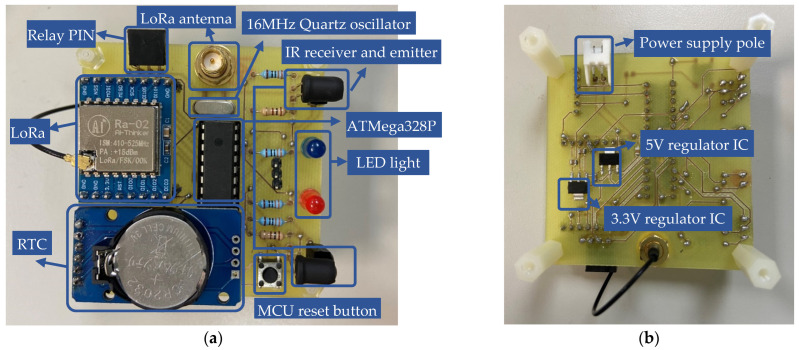
Physical diagram of the reflective-type sensor node: (**a**) **top** view; (**b**) **bottom** view.

**Figure 9 sensors-23-04185-f009:**
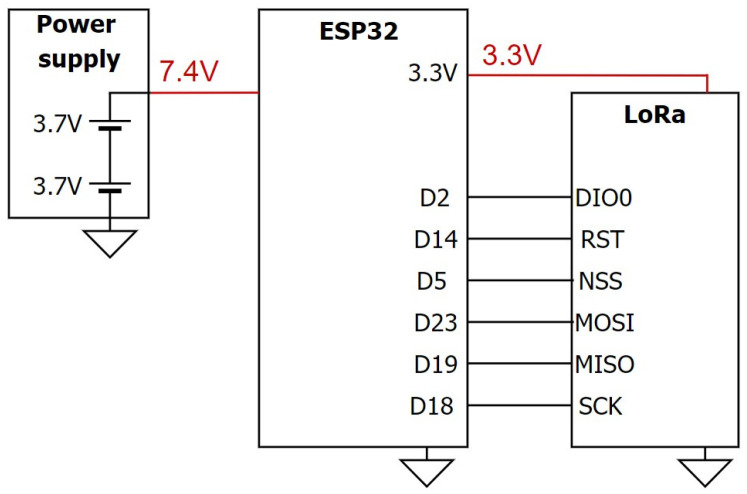
Block diagram of the proposed data collection center.

**Figure 10 sensors-23-04185-f010:**
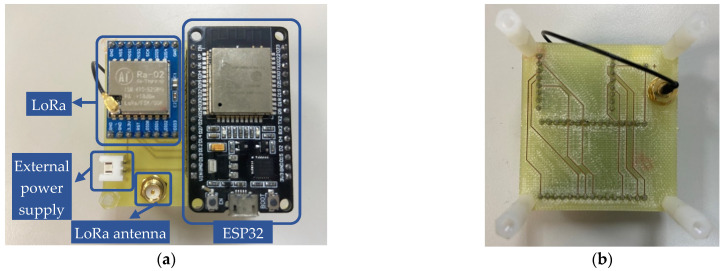
Physical diagram of the reflective-type sensor node: (**a**) **top** view; (**b**) **bottom** view.

**Figure 11 sensors-23-04185-f011:**
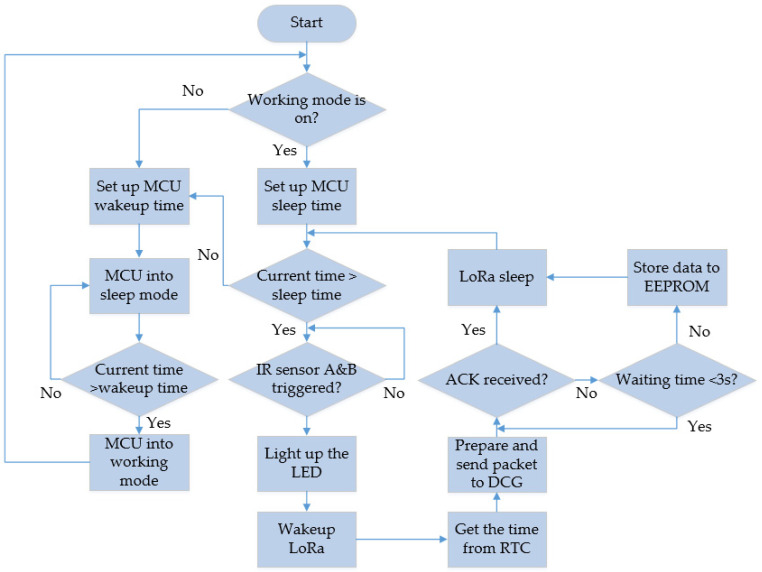
Flowchart of sensor modules.

**Figure 12 sensors-23-04185-f012:**
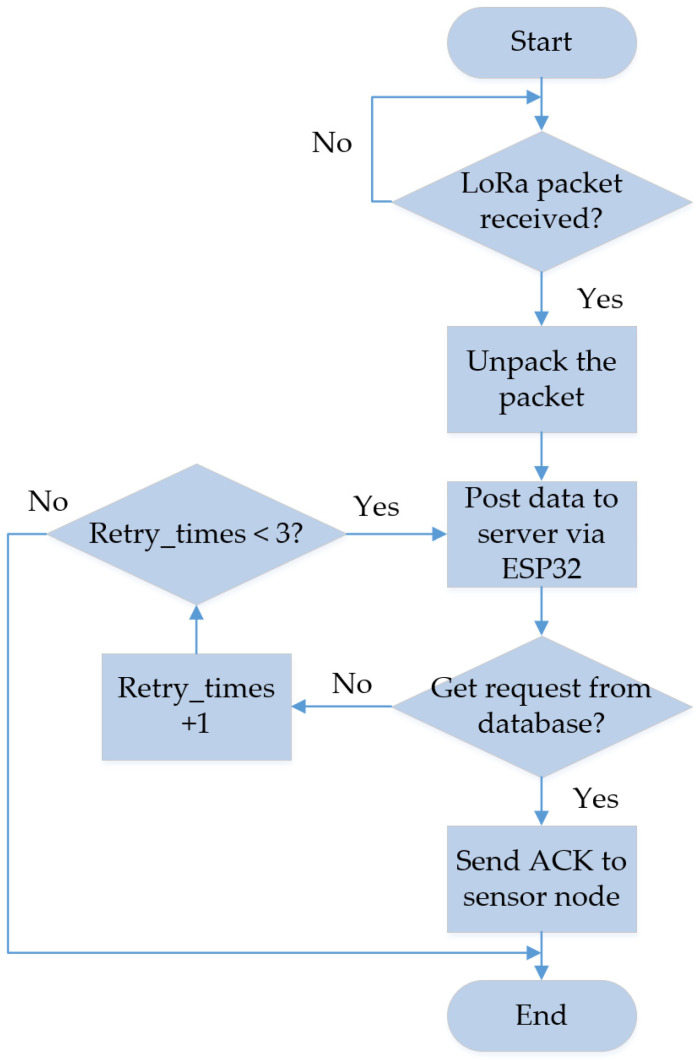
Flowchart for the DCG module.

**Figure 13 sensors-23-04185-f013:**
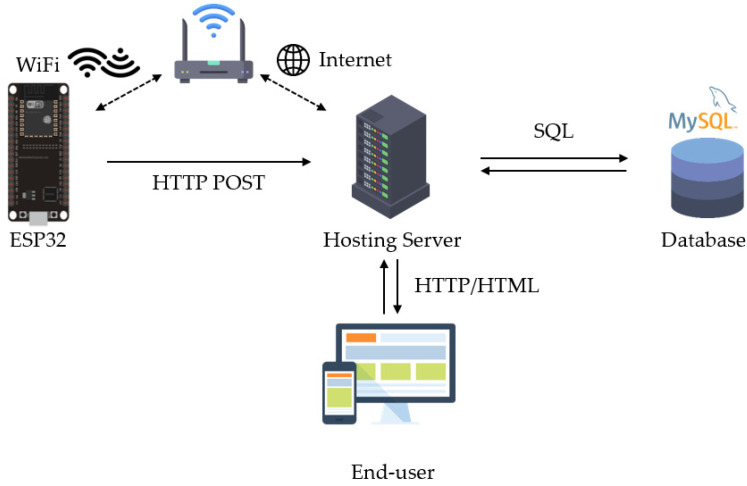
Overview of the software architecture.

**Figure 14 sensors-23-04185-f014:**
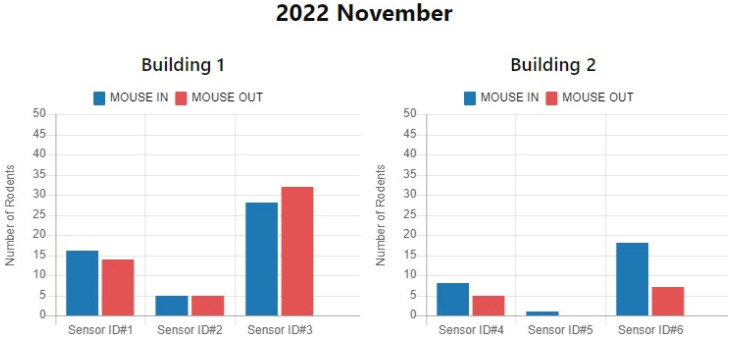
Rodent infestation data shown in the monitoring dashboard.

**Figure 15 sensors-23-04185-f015:**
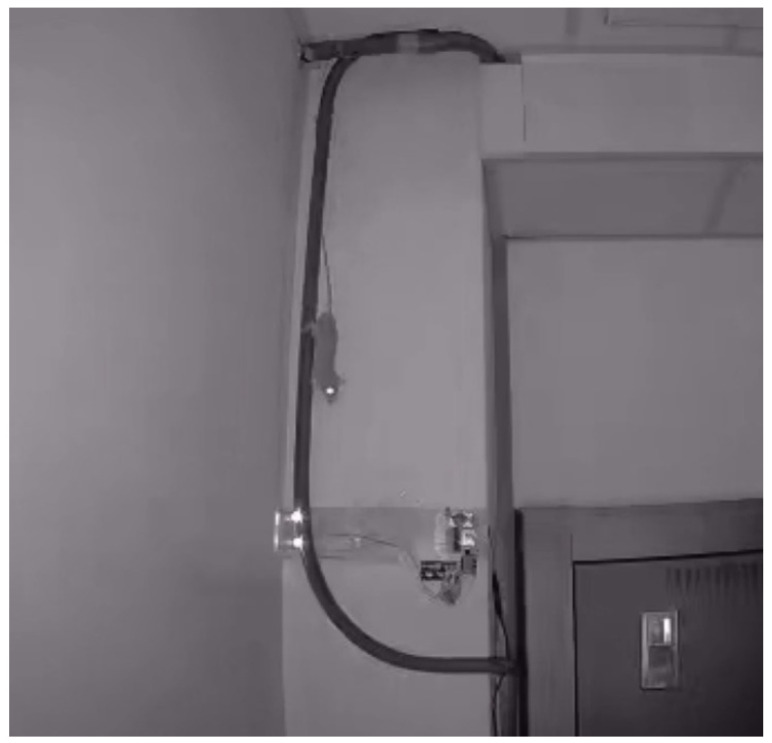
Captured image of the rat going through the sensing area of the sensor module.

**Figure 16 sensors-23-04185-f016:**
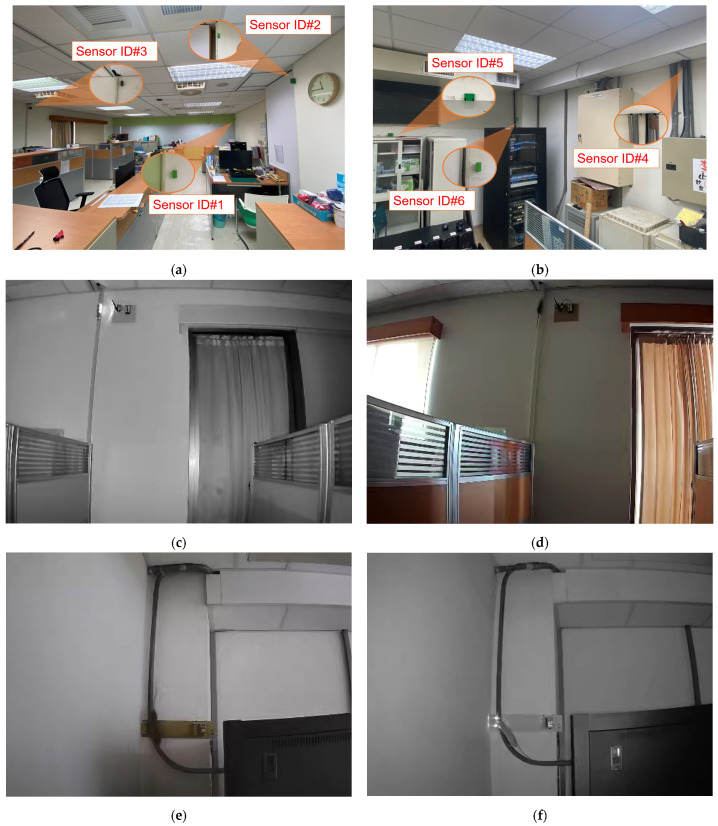
Setup and detection photos of rodents’ activity in different buildings: (**a**) sensor setup position in building 1; (**b**) sensor setup position in building 2; (**c**) mouse forward action detected in building 1; (**d**) mouse backward action detected in building 1; (**e**) mouse forward action detected in building 2; (**f**) mouse backward action detected in building 2.

**Table 1 sensors-23-04185-t001:** Comparison table of wireless communication technology.

Technology	Zigbee	Bluetooth	WiFi	Cellular	LoRa
Standard	IEEE 802.15.4	IEEE 802.15.1	IEEE 802.15.11 ah	3GPP	IEEE 802.15g
Network Type	Mesh	P2P	WLAN	GERAN	LPWAN
Spectrum	2.4 GHz	2.4 GHz	2.4–5 GHz	700–2600 MHz	433,868–915,923 MHz
Data Rate	0.25 Mbps	1 Mbps	5 Gbps	0.1–1 Gbps	250 Kbps
Max. Coverage	80 m	10 m	100 m	30 km	10 km

**Table 3 sensors-23-04185-t003:** Average Power Consumption for the Sensor Module.

Module Status	Average Current	Ratio of Daily PC	Time Period
Sensing Mode	37.8 mA	33.33% (8 h per day)	8 p.m.–4 a.m.
Communicating Mode	86.8 mA	0.036% (30 s per day)	8 p.m.–4 a.m.
Sleeping Mode	2 mA	66.66% (16 h per day)	4 a.m.–8 p.m.
Total Average Current	37.8 × 33.33% + 86.8 × 0.036% + 2 × 66.66% = 13.96 mA		

**Table 4 sensors-23-04185-t004:** The Setting Parameters of the LoRa Module.

Parameter	Value
Center frequency	433 MHz
Bandwidth	250 KHz
Coding rate	4/5
Spreading factor (SF)	12
Output power	14 dBm

**Table 5 sensors-23-04185-t005:** Cost of Each Sensor Module.

Item	Amount	Cost (USD)
Outer mold	1	2
LoRa Module	1	4.8
LoRa Antenna	1	3
Microcontroller	1	4
IR Emitter	2	0.2
IR Receiver	2	1
Batteries (18,650)	2	25
Miscellaneous	-	3
Total	-	43

**Table 6 sensors-23-04185-t006:** Cost of Each DCG Module.

Item	Amount	Cost (USD)
Outer mold	1	2
LoRa Module	1	5.5
LoRa Antenna	1	3
Microcontroller	1	6.5
Total	-	17

## Data Availability

Not applicable.

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
