# Peer review of "A Remote Monitoring System for Rodent Infestation Based on LoRaWAN"

_sensors, 2023, doi:10.3390/s23094185_

Round 1
Reviewer 1 Report
See attached!

Reviewer 2 Report
The following are a few suggestions/comments:
1. Should the Max. Coverage of Cellular in Table1 be 30 Km instead of 300m?
2. Regarding the possibility to add an ESP camera to your scheme in figure 3, to what extent may the overheads of the proposed scheme such as system setup, data storage, computing efficiency, cost, etc. be altered – more discussion could be added
3. In figure 14
a. please change Noverber to November.
b. Add the vertical axis measured variables (Number of Rodents).
4. Adding more discussion in the conclusions regarding the experimental results, to demonstrate the effectiveness of the proposed scheme and how your approach is acceptable could be useful. You may address a specific period of time, measuring the number of detected rodents while referring to figure 14.
5. What are the shortcomings of your proposed system? More discussion could be added.
6. Computation complexity, and security verification or validation of data could be added.
Reviewer 3 Report
The author should address the following: Why is your idea required? What difficulties are there? What remedies already exist for the issue you're trying to resolve? What are their disadvantages and restrictions?
The introduction could be expanded, and more major research sources should be cited.
The extended form of each acronym should be used when it is first used in the text.
Broad specifications and the benefits of utilizing a LoRaWAN network should be provided by the author.
The suggested system's accuracy and its parameters should be discussed by the author.
In the concluding part, it is important to underline the study's limitations as well as potential future steps in research.
Round 2
Reviewer 1 Report
The authors have made significant improvements to the paper.
I am uncomfortable with the Authors's response, “We hope you can appreciate our current predicament as we are unable to repeat the experiment due to the time constraints of the revised paper submission process.” on [Reply #4]. For future reference, the revised paper submission process should be flexible and not be used as an excuse not to revise the paper.
I look forward to the future paper on this project.
Authors may consider looking at the following literature to support the background information on LoRAWAN Performance:
https://www.mdpi.com/2079-9292/10/18/2224
https://www.mdpi.com/1424-8220/20/17/4670
https://ieeexplore.ieee.org/abstract/document/8703779
